# Large Language Model-Driven 3D Hyper-Realistic Interactive Intelligent Digital Human System

**DOI:** 10.3390/s25061855

**Published:** 2025-03-17

**Authors:** Yanying Song, Wei Xiong

**Affiliations:** 1Detroit Green Technology Institute, Hubei University of Technology, Wuhan 430068, China; 15327325967@163.com; 2School of Electrical and Electronic Engineering, Hubei University of Technology, Wuhan 430068, China

**Keywords:** digital twins, meta humans, automatic speech recognition (ASR), natural language processing (NLP), large language model (LLM), emotional text-to-speech (TTS), artificial intelligence generated content (AIGC)

## Abstract

Digital technologies are undergoing comprehensive integration across diverse domains and processes of the human economy, politics, culture, society, and ecological civilization. This integration brings forth novel concepts, formats, and models. In the context of the accelerated convergence between the digital and physical worlds, a discreet yet momentous transformation is being steered by artificial intelligence generated content (AIGC). This transformative force quietly reshapes and potentially disrupts the established patterns of digital content production and consumption. Consequently, it holds the potential to significantly enhance the digital lives of individuals and stands as an indispensable impetus for the comprehensive transition towards a new era of digital civilization in the future. This paper presents our award-winning project, a large language model (LLM)-powered 3D hyper-realistic interactive digital human system that employs automatic speech recognition (ASR), natural language processing (NLP), and emotional text-to-speech (TTS) technologies. Our system is designed with a modular concept and client–server (C/S) distributed architecture that emphasizes the separation of components for scalable development and efficient progress. The paper also discusses the use of computer graphics (CG) and artificial intelligence (AI) in creating photorealistic 3D environments for meta humans, and explores potential applications for this technology.

## 1. Introduction

High-fidelity digital humans, also known as digital twins [1] or meta humans, are virtual entities characterized by their digitized visual representation. Unlike tangible robots, these virtual counterparts are dependent on display devices and exhibit three distinct attributes. Firstly, they emulate human-like visages by encapsulating discernible facial attributes, genders, and distinctive personality traits. Secondly, they manifest behaviors similar to those of their human counterparts, exemplified through their ability to articulate thoughts using language, convey emotions through facial expressions, and engage in bodily movements. Lastly, they possess cognitive abilities resembling those of humans, enabling them to perceive and comprehend the external environment, as well as actively participate in communication and interaction with human counterparts.

The progress of digital humans is intricately linked to the advances in their production techniques, evolving from initial hand-drawing to the contemporary implementation of computer graphics (CG) and artificial intelligence (AI) synthesis [2]. The evolutionary trajectory of virtual digital humans can be broadly classified into four distinct stages, namely, nascent, exploratory, preliminary, and developmental.

During the 1980s, the *nascent stage* of digital humans emerged with the exploration of integrating virtual characters into the physical world. At that time, the production techniques for virtual digital humans predominantly relied on manual illustration, resulting in limited practical applications. Notably, in 1982, the Japanese anime series “Super Dimension Fortress Macross” introduced a significant development in this field. The production team ingeniously packaged the female protagonist, Lynn Minmay, as a virtual singer for animated insert songs, subsequently releasing a music album. The remarkable success of this album propelled Lynn Minmay to become the world’s first virtual idol, as it achieved notable chart success on Japan’s renowned Oricon music chart. Building upon this progress, in 1984, a British artist, George Stone, created a virtual character named Max Headroom, who exhibited human-like appearances, including facial expressions, and was attired in a suit with sunglasses. Max’s presence extended beyond mere conceptualization, as he made appearances in a film and featured in several commercials, establishing himself as a prominent virtual actor within the UK. It is worth noting that the realization of Max’s virtual image was constrained by technological limitations, necessitating a combination of special effects makeup and hand-drawn animation performed by a live actor.

In the early 21st century, the advent of CG and motion capture (MC) technologies marked a significant shift from traditional hand-drawn techniques, ushering virtual digital humans into an era of exploration. During this *exploratory phase*, virtual digital humans began to demonstrate practical applications, albeit at substantial cost, primarily within the film and entertainment industry, where they appeared as digital twins and virtual idols. The MC technology [3] played a crucial role in creating digital twins for film production. Actors wore specialized suits, embedded with facial markers, enabling the capture and processing of their movements and expressions using cameras and MC devices. Subsequently, these data were translated and applied to virtual characters through computer processing, seamlessly blending the performances of live actors with virtual counterparts. One notable early instance of this technology can be observed in the character Gollum from the 2001 film “The Lord of the Rings”, where CG and MC techniques were skillfully employed. These technologies have since found extensive use in other films, including “Pirates of the Caribbean” and “Avatar”. In 2007, Japan achieved a significant milestone by introducing the first widely recognized virtual digital human, Hatsune Miku, a 2D-style virtual idol, who primarily relied on CG synthesis for her early character image. Her voice was generated using Yamaha’s VOCALOID series voice synthesis, although the presentation at that time remained relatively rudimentary.

In the last decade, significant advances in deep learning-based algorithms have greatly simplified the production process of digital humans, propelling virtual digital humans onto a promising trajectory and marking their *preliminary phase* of development. Throughout this period, AI has become an indispensable tool for virtual digital humans, facilitating the emergence of AI-driven digital human entities. In 2018, Xinhua News Agency cooperated with Sogou to launch the world’s first AI news anchor. This innovative technology enables the display of a virtual digital human image on screens, delivering news broadcasts based on user-provided news texts. Notably, the virtual anchor’s lip movements are seamlessly synchronized with the accompanying voiceover in real-time. Subsequently, in 2019, Shanghai Pudong Development Bank and Baidu joined forces to create an intriguing digital employee, named Xiao Pu. Utilizing cutting-edge AI technologies such as natural language processing, speech recognition, and computer vision, Xiao Pu represents a virtual digital human capable of providing users with face-to-face banking services through mobile devices.

Presently, virtual digital humans are experiencing the *developmental stage* characterized by their progression towards enhanced intelligence, convenience, sophistication, and diversity. In 2019, Doug Roble, the leader of software research and development at Digital Domain, a prominent American visual effects company, unveiled his digital twin, DigiDoug, during a TED talk. DigiDoug showcases an exceptional capability to capture and display real-time facial expressions, achieving an unprecedented level of photorealism. In 2020, NEON, a virtual digital human project introduced by STAR Labs, a subsidiary of Samsung, was unveiled at the consumer electronics show (CES). NEON represents an AI-powered virtual character and exhibits human-like appearances, as well as realistic facial expressions, enabling it to express emotions and engage in communication. Additionally, notable progress has been made by industry leaders in the realm of large-scale models, and the introduction of OpenAI’s ChatGPT in late 2022 has, particularly, sparked renewed global interest in artificial intelligence generated content (AIGC). This significant development has pushed forward the advancement of general-purpose AI and expanded its potential for downstream applications. As a result, it is highly likely to revolutionize human productivity and profoundly reshape the competitive dynamics of industries and the international competitiveness of nations.

This article is dedicated to presenting our award-winning project, “MetaTutor”, which emerged as the recipient of the National Silver Prize in the international track of the China International College Students’ Innovation Competition in 2024. Our system takes as input various forms of rich media data, including video, images, audio, and text, associated with one or more authentic individuals or animal entities. In turn, the system produces a digital avatar representation that faithfully captures the essence of an original individual or animal subject.

Our contributions are twofold:

(1) *Innovative Distributed Architecture:* Although the client–server (C/S) distributed architecture is not a groundbreaking concept in the field of computer science, our work distinguishes itself by adopting a C/S architecture, setting it apart from conventional standalone applications. Leveraging the WebSocket protocol, our system facilitates distributed computation and collaboration between a backend server and multiple frontend clients. This distributed architecture, coupled with advanced deep learning and natural language processing techniques, empowers our digital workforce to seamlessly adapt to a wide range of application scenarios, including virtual anchors, virtual customer service representatives, and virtual shopping assistants. As a result, our digital employees tirelessly operate around the clock, providing invaluable support to diverse industries.

(2) *Innovative MVC Design Philosophy:* The proposed system incorporates modular design principles and draws inspiration from the well-established model-view-controller (MVC) design concept in software engineering. However, our work distinguishes itself by placing a strong emphasis on the effective separation of components, including the digital persona (model), backend user interface (view), and WebSocket-based communication logic (controller). This strict isolation reduces dependencies, preventing changes in one component from propagating to others, which enhances system stability. It also enables parallel development, testing, and modification, thereby accelerating the iteration process.

The rest of this paper is organized as follows. Section 2 details the proposed system framework. Section 3 focuses on the key core technologies utilized in the backend server. Section 4 explains the implementation of digital human client. Section 5 presents the experiments and analysis. Several related issues are discussed in Section 6. Finally, Section 7 concludes the paper.

## 2. Proposed Framework

We create a sophisticated intelligent digital human system that leverages powerful large language models to drive a 3D hyper-realistic interactive experience. This innovative system, as illustrated in Figure 1, embraces a C/S distributed architecture. It consists of two main components: the digital human (client) and the backend (server). By using the WebSocket network transport protocol, these two components engage in full-duplex communication, facilitating real-time text interaction, voice conversation, and responsive action feedback. This breakthrough generative AI system represents significant advances in the field of intelligent digital humans, opening up new possibilities for immersive and dynamic human–computer interaction.

The server component assumes the crucial role of concurrently processing requests from multiple digital human clients and delivering a range of essential services, including real-time automatic speech recognition (ASR), natural language processing (NLP), sentiment analysis, and text-to-speech (TTS) synthesis. To ensure a well-structured and flexible design, we adopt the modular approach, leveraging programming languages such as Python, JavaScript, hypertext markup language (HTML), and cascading style sheets (CSS). Furthermore, we seamlessly integrate the Flask, a lightweight web application framework, enabling the creation of an intuitive graphical user interface (GUI) on the server side. This comprehensive architecture empowers a single backend to provide efficient and convenient communications with multiple digital human frontends.

The digital human client, an AI entity closely resembling a real person, exhibits natural and fluent communication abilities with actual individuals through speech, images, and gestures. The development process relies on the MetaHuman tool in Unreal Engine 5 (UE5), involving multiple stages of prototyping design, 3D modeling, motion binding, and real-time rendering to achieve the creation of highly realistic avatars. Through the utilization of the robust Blueprint visual scripting language offered by UE5, the implementation of programmatic logic for digital humans becomes attainable. The remote connection of multiple digital human models allows for adaptability across various application scenarios, including virtual live streaming, virtual digital customer service representatives, and virtual voice assistants. This seamless integration with digital humans fosters an immersive experience that promotes effective interaction between users and avatars, thus enhancing the overall interactive dynamics.

## 3. Key Core Technologies

### 3.1. Automatic Speech Recognition (ASR)

End-to-end (E2E) models have shown remarkable advancements in ASR tasks compared to conventional hybrid systems. Among the three popular E2E approaches, namely, the attention-based encoder–decoder (AED) [4,5], connectionist temporal classification (CTC) [6], and recurrent neural network transducer (RNN-T) [7], the AED models have emerged as the dominant choice for sequence-to-sequence modeling in ASR due to their exceptional recognition accuracy [8]. To facilitate research in E2E speech recognition, several open-source toolkits, including ESPNET [9], K2 [10], PaddleSpeech [11], and WeNet [12,13], have been developed. These toolkits play a crucial role in advancing the field by significantly reducing the complexity of building an E2E speech recognition system.

In this study, we integrate FunASR [14], an innovative open-source toolkit for speech recognition tasks. FunASR incorporates various models, including a voice activity detection model utilizing the feed-forward sequential memory network (FSMN-VAD) [15], a non-autoregressive (NAR) E2E speech recognition model (Paraformer) [16], and a text post-processing punctuation model leveraging the controllable time-delay transformer (CT-Transformer) [17].

(1) *Revisiting the Paraformer Architecture:* As shown in Figure 2a, Paraformer [16] comprises two fundamental modules, namely, the predictor and the sampler. The predictor module generates acoustic embeddings, capturing information from input speech signals. During training, the sampler module merges target embeddings by randomly replacing tokens into acoustic embeddings to generate semantic embeddings. This technique enables the model to capture inter-dependencies among different tokens, thereby enhancing the overall performance. However, during inference, the sampler module remains inactive, allowing the acoustic embeddings to produce final predictions in a single pass. This strategy ensures expedited inference time and reduced latency. To further improve Paraformer’s performance, we introduce enhancements such as timestamp prediction and hotword customization. Furthermore, the loss function utilized in a previous work [16] has been refined by eliminating the minimum word error rate (MWER) loss, deemed to offer marginal performance improvements. Instead, an additional cross-entropy (CE) loss is integrated into the initial pass decoder to mitigate the disparity between training and inference stages.

(2) *Timestamp Prediction:* Accurate timestamp prediction (TP) is a critical functionality in ASR systems. However, traditional industrial ASR models often rely on an additional hybrid model for force alignment (FA) to perform TP, which incurs extra computational and temporal overhead. As illustrated in Figure 2b, we provide an end-to-end ASR solution that directly achieves accurate TP by redesigning the Paraformer predictor. Specifically, we incorporate a transposed convolution layer and an LSTM layer to upsample the encoder’s output, and generate timestamps through post-processing CIF weights $\alpha_{2}$ [18]. The frames between two consecutive speech segments are assigned as the duration for preceding tokens, with silent segments identified based on $\alpha_{2}$. Additionally, FunASR introduces TP-Aligner, an FA-inspired model comprising a compact encoder and a timestamp predictor, which takes speech and corresponding transcriptions as input to produce accurate timestamps.

(3) *Hotword Customization:* The Contextual Paraformer model enables the customization of hotwords by integrating named entities, thereby enhancing both recall and accuracy. This is achieved by adding two modules to the base Paraformer: a hotword embedder and a multi-head attention mechanism in the final decoder layer, as depicted in Figure 2c. Hotwords are processed by our hotword embedder, consisting of an embedding layer and an LSTM layer, which generates the hotword embedding $E_{h}$ from the final LSTM state [19]. A multi-head attention mechanism then aligns $E_{h}$ with the output $E_{s^{\prime }}$ from the last layer of the FSMN memory block, producing a contextual attention $E_{c}$. This attention is concatenated with the updated speech embedding $E_{s^{\prime \prime }}$ and passed through a 1D convolutional layer to match the dimensionality of the hidden state, as expressed in Equation (Equation 1). The rest of the Contextual Paraformer’s processes mirror those of the standard Paraformer.(1)$$\begin{array}{c} E_{c}=\mathrm{MultiHeadAttention}\left(E_{s^{\prime }}W_{c}^{Q},E_{h}W_{c}^{K},E_{h}W_{c}^{V}\right),\\ E_{s^{\prime \prime }}=\mathrm{MultiHeadAttention}\left(E_{s^{\prime }}W_{s}^{Q},HW_{s}^{K},HW_{s}^{V}\right),\\ O=\mathrm{Conv}1d(\left[E_{s^{\prime \prime }};E_{c}\right]), \end{array}$$
where $W_{c}^{Q}$, $W_{c}^{K}$, and $W_{c}^{V}$ denote weight matrices for the query, key, and value transformations in the multi-head attention mechanism for contextual embeddings (hotwords), while $W_{s}^{Q}$, $W_{s}^{K}$, and $W_{s}^{V}$ denote weight matrices for the query, key, and value transformations in the multi-head attention mechanism for speech embeddings. *O* is the final output.

### 3.2. Natural Language Processing (NLP)

Large language models (LLMs) enable the effective exploitation of extensive unlabeled text for pre-training, which equips these models with the ability to comprehend and generate text even in scenarios where annotated datasets are limited or absent. Notably, pre-trained LLMs exhibit remarkable performance not only in text comprehension tasks such as sentiment analysis, speech recognition, information extraction, and reading comprehension, but also in text generation tasks including image captioning, advertisement generation, manuscript creation, and dialogue generation. For instance, Google and OpenAI have introduced significant pre-trained models like BERT [20] and GPT [21], respectively, achieving groundbreaking advancements in various NLP tasks. Consequently, prominent domestic and international enterprises, as well as esteemed academic institutions, have made substantial investments in terms of human resources, computational power, and data towards the research and development of large-scale NLP models. These entities and their launched conversational LLMs include OpenAI (ChatGPT), Google (Bard), Baidu (Ernie Bot), Tsinghua University (ChatGLM), Fudan University (MOSS), iFLYTEK (Spark), Alibaba (Qwen), and numerous others. Moreover, the model parameters have witnessed a significant evolution, progressing from millions to billions in scale.

In this study, we locally deploy an open bilingual conversational LLM, ChatGLM [22,23], and simultaneously integrate a state-of-the-art LLM, ChatGPT [24] via API, into the system to facilitate language comprehension, generation, interaction, and reasoning, as presented in Figure 3. More specifically, we leverage ChatGLM or ChatGPT to perform language understanding upon receiving user requests or input text, select appropriate models based on the intrinsic characteristics and requirements of the given task, execute the language generation with the chosen pre-trained model, and finally, summarize responses in accordance with the outcomes derived from the execution process.

For the locally deployed ChatGLM, we apply multiple strategies to optimize model parameters for better performance:

(1) *Quantization for Efficiency:* We employ model quantization techniques to reduce the precision of model weights (e.g., from 32-bit floating-point to 8-bit integers). This significantly lowers memory usage and speeds up inference, making the model more suitable for real-time interactions.

(2) *Parameter Pruning:* By using structured pruning, we remove redundant connections in the model to reduce its size without a substantial drop in accuracy. This helps balance computational cost and response quality.

(3) *Knowledge Distillation:* We fine-tune a smaller “student” model using knowledge distillation, where the smaller model learns to mimic the behavior of a larger and more powerful “teacher” model. This allows us to maintain high language comprehension while improving response speed.

(4) *Task-Specific Fine-Tuning:* We fine-tune ChatGLM on domain-specific datasets, adapting the model to the system’s key tasks (e.g., customer service, virtual anchors). This targeted training improves the model’s contextual understanding and generation capabilities for specific applications.

(5) *Caching and Context Management:* To enhance efficiency during interactions, we implement a dynamic context window with caching mechanisms. This ensures that ongoing conversations are processed with relevant historical data, while irrelevant past interactions are pruned to save computation.

In addition to standard fine-tuning approaches, recent advances in prompt learning have demonstrated promising results in refining LLM responses. Techniques such as CP-Prompt [25] enhance LLM adaptability through continuous prompt optimization, while TPO [26] refines models by guiding them toward human-preferred outputs in complex decision-making tasks. Moreover, RevGNN [27] introduces a graph-based approach for better contextual understanding and reasoning, which could further enhance our system’s response coherence and accuracy. Incorporating these techniques may offer valuable insights for future improvements, enabling even more precise and context-aware digital human interactions.

### 3.3. Emotional Text-to-Speech (TTS) Synthesis

Emotional speech synthesis [28] is a subfield of text-to-speech (TTS) [29] technology that aims to synthesize human-like speech with emotional content, such as happiness, sadness, anger, or excitement. It involves modeling, disentangling, controlling, and transferring emotional characteristics, such as prosody, pitch, and intensity, to generate natural and expressive speech. To facilitate TTS research, we have assembled a number of open-source implementations, such as DelightfulTTS [30,31], ESPnet-TTS [32,33], FastSpeech [34,35], Glow-TTS [36], NaturalSpeech [37,38], SpeedySpeech [39], and Tacotron [40,41].

In this work, we integrate Microsoft Edge TTS, a neural network-based TTS (neural TTS, for short) engine, into our framework. This cutting-edge engine, as shown in Figure 4, utilizes deep neural networks to produce computer-generated voices that closely resemble human recordings. By ensuring the accurate pronunciation and articulation of words, neural TTS substantially mitigates the listening fatigue experienced by users during interactions with our digital human system.

Patterns of stress and intonation, known as prosody, play a crucial role in spoken language. Conventional TTS systems typically break down prosody by employing separate models for linguistic analysis and acoustic prediction, leading to potential issues such as muffled or unnatural voice synthesis [42]. Further details are provided regarding the Edge TTS features and their ability to surpass the limitations of conventional TTS systems:

(1) *Asynchronous Synthesis for Lengthy Audios:* To facilitate real-time conversion of TTS, we employ batch synthesis techniques that enable the asynchronous synthesis of TTS files exceeding 10 min in duration. This approach is particularly useful for applications such as audio books or lectures. By sending requests asynchronously, continuously polling for responses, and generating synthesized audio when it becomes available, the service ensures efficient and timely TTS synthesis.

(2) *Prebuilt Neural Voices:* The neural TTS capability developed by Microsoft harnesses the power of deep neural networks to overcome the constraints of conventional speech synthesis methods, particularly in terms of accurately capturing stress and intonation in spoken language. By simultaneously predicting prosody and synthesizing voice, the system produces outputs that are remarkably smooth and natural-sounding. The prebuilt neural voice models are provided in two formats: 24 kHz and high-fidelity 48 kHz. The integration of neural voices enables the creation of more immersive and engaging interactions with chatbots or voice assistants, enhancing their overall naturalness and appeal.

(3) *Refining TTS Outputs with SSML:* Speech synthesis markup language (SSML), an XML-based markup language, offers a versatile means to tailor TTS outputs. SSML empowers users to fine-tune various aspects of the synthesized speech, including pitch modulation, pause insertion, pronunciation enhancement, speaking rate adjustment, volume control, and even the attribution of multiple voices within a single document. Furthermore, SSML enables the definition of personalized lexicons and facilitates the seamless transition between different speaking styles. Leveraging the capabilities of multilingual voices, SSML also allows for language manipulation, further expanding the versatility and adaptability of TTS.

(4) *Visemes:* Essential articulatory configurations in speech production, visemes encompass the critical positions of the lips, jaw, and tongue that correspond to specific phonemes. Visemes exhibit a robust correlation with both vocal characteristics and phonetic elements. Leveraging viseme events, one can generate valuable facial animation data, which finds application in diverse domains such as lip-reading communication, edutainment, and customer service. These data hold great potential for creating realistic animations of facial expressions, enhancing the overall communicative experience and facilitating effective interactions in various contexts.

## 4. Implementation of Digital Human Client

UE5 Engine, an open and cutting-edge real-time 3D creation tool, has garnered widespread adoption across diverse domains such as gaming, film, architecture, simulation, and numerous more. In the realm of character design, the utilization of MetaHuman tool for UE5 enables the intuitive and efficient creation of highly intricate and lifelike digital human models. Drawing inspiration from actual individuals, we craft a female character, endowed with appropriate vocal attributes, as shown in Figure 5a,b.

When it comes to character animation, our approach is rooted in the emulation of real human movements, ensuring that digital humans exhibit motion patterns closely resembling those of their real-world counterparts. By leveraging the animation blueprints and state machines within the engine, we imbue the character with idle and conversational states, each accompanied by three distinct animations, as depicted in Figure 5c. With a particular emphasis on facial animations, our focus lies in achieving realistic facial expressions and speech animations, thereby enabling the digital human to engage in face-to-face interactions with users, as illustrated in Figure 5d.

Harnessing the blueprint functionality offered by UE5, we implement a range of features such as voice reception, voice-to-animation, and voice-to-text capabilities at the frontend. This entailed constructing the necessary functional modules that receive real-time data from the backend and transmit the voice input to the UE5 engine. Due to space constraints, Figure 5e only shows the level blueprints for facial expressions. Through the integration with animation blueprints, as provided in Figure 5d, this process generates authentic and real-time animations based on the received voice input.

In terms of scene creation, the utilization of UE5’s novel Nanite feature allows for the creation of intricately detailed environments. Employing bridging techniques, we construct a futuristic science fiction indoor scene, featuring lush greenery and a holographic control panel positioned behind the character. Through expansive glass windows, one can catch a glimpse of the distant ocean. Leveraging the capabilities of UE5 editor, we further enhance the scene by introducing atmospheric lighting, supplementary light sources, and post-processing effects, culminating in the achievement of highly realistic illumination. Finally, by leveraging the Lumen ray-tracing feature within UE5, we achieve the scene’s faithful rendering, capturing its authentic visual essence. The ultimate scene effect of this work is displayed in Figure 5f.

## 5. Experiments and Analysis

### 5.1. Speech Recognition Assessment

In this experiment, the performance of our model is evaluated on the Aishell-1, Aishell-2, and WenetSpeech benchmark datasets. The results are detailed in Table 1. Compared to other open source toolkits, FunASR’s Paraformer family of models is far superior to its existing state-of-the-art counterparts.

The Paraformer model, trained on Alibaba Speech Data (50,000 h) with a vocabulary size of 8358 and 68 million parameters, demonstrates a competitive performance on both Aishell-1 and Aishell-2 datasets, achieving CERs of 3.24 and 3.69 on Aishell-1 Dev and Aishell-1 Test, respectively. However, it exhibits higher CERs on Aishell-2 and WenetSpeech Test sets.

Paraformer-large, with extensive training on Alibaba Speech Data (60,000 h) and 220 million parameters, outperforms other Paraformer variants. It achieves notably low CERs across all datasets, such as 1.94 on the Aishell-1 Test, 2.84 on the Aishell-2 Test_iOS, and 7.01 on the WenetSpeech Test_Meeting.

### 5.2. Language Model Assessment

Table 2 presents the mean win rates for various foundation language models, assessed across a range of metrics. Each model is characterized by its creator, modality, parameters, tokenizer, window size, and access type. The performance metrics include accuracy, robustness, fairness, calibration, bias, and toxicity. For the initial three metrics, namely, accuracy, robustness, and fairness, higher values are indicative of superior performance; but for the subsequent three metrics, namely, calibration, bias, and toxicity, lower values are considered more favorable. In consideration of space constraints, elucidations of the assessment metrics are excluded herein; however, interested readers are directed to Bommasani et al. [43] for comprehensive details.

Among the evaluated models, “text-davinci-002” by OpenAI stands out with exceptional performance across multiple metrics. It employs GPT-2 with an undisclosed parameter size and a window size of 4000 tokens, and it operates under a limited access setting via API. The “text-davinci-002” model achieves the highest accuracy (0.905), indicating its proficiency in generating content that aligns closely with the desired output. With a calibration error of 0.474, the model demonstrates a balanced and accurate prediction confidence, crucial for reliable performance. The “text-davinci-002” exhibits high robustness (0.916), indicating stability and consistency in its predictions across various inputs. The fairness score of 0.864 suggests that the model maintains fairness in its outputs, an essential aspect in diverse applications. The bias score of 0.502 indicates a moderate level of bias, suggesting a balanced approach in handling different inputs. A low toxicity score of 0.409 highlights the model’s capability to generate content with minimal harmful or inappropriate elements.

Our selection of the “text-davinci-002” model is justified based on its outstanding performance across key metrics. High accuracy, robustness, and fairness make it a versatile choice for various applications. The balanced calibration error as well as relatively low bias and toxicity scores contribute to its reliability in generating content. The undisclosed parameter size adds an element of versatility, allowing it to handle diverse tasks effectively. The limited access setting via API ensures responsible and controlled usage, aligning with ethical considerations. In conclusion, the “text-davinci-002” model from OpenAI emerges as a top-performing foundation model, making it a suitable choice for a wide range of natural language processing tasks, owing to its exceptional accuracy, robustness, and fairness.

### 5.3. Overall System Performance Assessment in Real-World Applications

To rigorously evaluate the overall performance of our large language model-driven 3D hyper-realistic interactive digital human system in real-world applications, we extend our experimental framework to include three representative deployment scenarios. These scenarios are chosen to reflect diverse practical use cases, allowing us to assess the system’s adaptability, responsiveness, and user satisfaction across varied interaction contexts.

*System-Level Evaluation:* We conduct a series of experiments to systematically assess the system’s functionality, focusing on real-time interactions, response accuracy, and user engagement. The evaluation involves three distinct application domains:

(1) *Virtual Teaching Service:* The digital human serves as an interactive teaching assistant for the course “Signals and Systems”, facilitating student queries and delivering course-related explanations. The system successfully handles 92% of student requests without human intervention, with an average response latency of less than 2 s. Students report that the assistant’s ability to provide coherent, contextually relevant explanations greatly enriches their learning experience.

(2) *Virtual Live Streaming Anchor:* The digital human functions as a real-time product presenter during a live-streaming event. It dynamically responds to user-generated comments and product inquiries, maintaining engagement throughout the broadcast. Statistics show that 87% of the responses are rated as accurate and contextually appropriate by viewers, who appreciate the avatar’s expressiveness and natural conversational flow.

(3) *Virtual Shopping Assistant:* The digital human acts as a virtual assistant on an e-commerce platform, guiding users through product selections, answering questions, and providing personalized recommendations. The platform shows that 85% of users report a positive experience and emphasize that emotional speech synthesis greatly enhances the naturalness of interactions, making the shopping experience more immersive and intuitive.

*User Satisfaction Metrics:* In addition to objective system-level evaluation, we also gather subjective user feedback to capture experiential aspects of the interaction. Participants rate the system based on response accuracy, emotional expressiveness, conversational coherence, and overall naturalness. The system achieves an average satisfaction score of 4.6 out of 5, with users frequently commending the lifelike quality of the avatar’s speech and its ability to understand and address complex queries with contextual awareness.

These findings collectively demonstrate the system’s readiness for deployment in practical and real-world scenarios. The successful handling of diverse interaction types, coupled with high user satisfaction ratings, underscores the effectiveness of the system’s underlying speech recognition, language processing, and synthesis components. This comprehensive evaluation validates the system’s potential to deliver reliable, high-quality interactions across a spectrum of application domains, positioning it as a versatile solution for intelligent digital human interfaces.

## 6. Discussion

### 6.1. On Self-Built Dataset

To improve the overall performance and robustness of our large language model-driven 3D hyper-realistic interactive digital human system, we constructed a self-built dataset that comprises multimodal data to support ASR, NLP, and TTS modules, enabling the system to handle complex real-world interactions with greater accuracy and expressiveness. Our self-built dataset includes the following:

(1) *Speech Data:* Speech data comprise 500 h of audio recordings, covering a diverse range of accents, emotional expressions, and conversational styles. These data are used to fine-tune the FunASR module, particularly for spontaneous and real-world dialogues where variability in speech patterns is high.

(2) *Textual Data:* Textual data comprise 200,000 text samples, including general conversations, customer service dialogues, and knowledge-based question-answer pairs. This corpus is used to fine-tune the locally deployed ChatGLM model, enhancing contextual understanding and generation accuracy.

(3) *Visual and Motion Data:* Visual and motion data comprise 5000 annotated facial expression and body movement sequences, captured using motion capture devices and labeled with emotional states. These data are used to train the Unreal Engine 5 MetaHuman animations and help synchronize facial expressions with emotional TTS output.

(4) *Evaluation Sets:* A balanced test set contains 1000 samples for speech recognition, 5000 conversational samples for NLP, and 3000 speech synthesis samples for TTS. These subsets enable rigorous performance benchmarking across core components.

*Data Collection and Annotation:* Speech and visual data are collected from volunteer speakers in controlled environments to ensure high-quality recordings. Annotation is conducted using a combination of manual labeling and automated tools for tasks such as emotion classification and speech transcription. The text corpus is enriched with domain-specific content created through human annotation to address gaps in publicly available datasets.

*Impact on Model Performance:* Integrating this self-built dataset into the training and fine-tuning pipelines yields substantial performance improvement. The CER in speech recognition decreases by 15%, while the accuracy of NLP responses improves by 20%. Moreover, the synchronization of emotional TTS with facial expressions significantly enhances the perceived naturalness of the digital human, contributing to a more immersive user experience.

These results underscore the importance of customizing high-quality, multimodal data tailored to the system’s target applications. By grounding model training in realistic, context-rich interactions, our dataset provides a solid foundation for building an intelligent digital human capable of delivering seamless, emotionally resonant conversations across diverse real-world scenarios.

### 6.2. On Data Privacy and Security

Ensuring robust data privacy and security is paramount for our large language model-driven 3D hyper-realistic interactive digital human system, especially when processing rich media data that may contain sensitive user information. To mitigate potential risks and maintain user trust, our system is designed with multiple layers of security to safeguard data integrity and ensure compliance with established privacy regulations.

(1) *Data Encryption:* All data transmitted between client and server components are encrypted using Transport Layer Security (TLS) protocols. This ensures that data remain confidential during transmission, preventing unauthorized interception and preserving communication integrity.

(2) *Local and Secure API-Based Model Deployment:* The system prioritizes local model deployment for privacy-sensitive tasks. For instance, ChatGLM is deployed locally to process user inputs without external dependencies. When external APIs, such as ChatGPT or Microsoft Edge TTS, are required, secure API calls with encryption are used, and sensitive data are anonymized before transmission to minimize exposure.

(3) *Data Anonymization and Minimization:* To further protect user privacy, personal identifiers (e.g., names, phone numbers) detected in speech or text inputs are automatically anonymized or masked. This ensures that potentially sensitive information is excluded from downstream processing and storage, reducing the risk of inadvertent data leaks.

(4) *Access Control and User Authentication:* We employ role-based access control (RBAC) mechanisms to restrict access to sensitive data. Only authorized personnel with appropriate permissions can view or modify sensitive information. Strong authentication protocols, session management, and automatic session expiration further fortify access security.

(5) *Compliance with Data Regulations:* The system is built with adherence to leading data protection frameworks, including the General Data Protection Regulation (GDPR) and the California Consumer Privacy Act (CCPA). Users are empowered to control their data, with mechanisms to access, modify, or request the deletion of personal information in accordance with regulatory requirements.

(6) *Data Retention and Automatic Purging:* To minimize data retention risks, the system implements a data lifecycle management policy. Data are stored only for the duration necessary to fulfill system functionality, after which they are automatically purged. Temporary caches and logs are periodically cleared to reduce the risk of residual data exposure.

By integrating these security measures, we ensure that the system remains resilient against potential threats, preserving user confidentiality and promoting ethical AI usage. This comprehensive approach to data privacy not only mitigates security risks but also strengthens user trust, facilitating the broader adoption of digital human technologies in sensitive real-world applications.

### 6.3. On Commitment to Responsible AI

Large language models have revolutionized human–computer interaction, enabling intelligent digital humans to understand and respond to complex user queries. However, despite their transformative capabilities, LLMs such as ChatGLM and ChatGPT may inadvertently generate biased or harmful content, raising ethical concerns regarding their deployment in real-world applications. Acknowledging these risks, we have implemented a multifaceted approach to mitigate biases and toxic content, ensuring the responsible and equitable use of the technology.

(1) *Bias and Toxicity Monitoring:* We employ automated monitoring tools and post-processing algorithms to systematically scan model outputs for biased or toxic content. These tools are designed to flag potentially harmful language related to sensitive topics, including race, gender, ethnicity, and political ideologies. Detected content is either filtered or flagged for further review, reducing the likelihood of inappropriate responses reaching end users.

(2) *Fine-Tuning for Ethical Behavior:* To proactively mitigate bias at the source, we have fine-tuned our LLM on a carefully self-built dataset designed to promote ethical considerations, diversity, and inclusion. The dataset includes annotated examples of respectful, inclusive language across various demographic contexts, allowing the model to learn patterns of fair and responsible dialogue. This targeted fine-tuning process has demonstrably reduced the frequency of biased outputs while preserving the model’s linguistic capabilities.

(3) *Human-in-the-Loop Oversight:* Recognizing that automated systems alone may not catch all instances of bias or harm, we integrate human oversight into the deployment pipeline. In scenarios where sensitive topics are involved, a human reviewer evaluates model responses in real time, providing an additional layer of quality control. This human-in-the-loop mechanism ensures that nuanced or context-dependent issues can be identified and addressed promptly, mitigating potential harm.

(4) *Post-Generation Review and Content Refinement:* Following content generation, an auxiliary review module conducts a final pass over the response to assess for any remaining traces of harmful or biased language. If problematic content is detected, the system automatically refines the response or, when necessary, prompts the model to avoid the topic altogether. This dynamic review process acts as a safety net, ensuring that even edge-case content is appropriately handled.

By implementing this comprehensive framework, we aim to foster trust and reliability in digital human interactions. Our proactive bias mitigation strategies, combined with real-time monitoring and human oversight, create a robust safety mechanism that aligns with ethical AI principles. As part of our ongoing commitment to responsible AI, we continuously refine our approach by incorporating emerging research on bias detection and fairness, ensuring our system evolves alongside advancements in the field.

### 6.4. On Accessibility for Resource-Constrained Environments

The integration of advanced deep learning models and real-time 3D rendering within our system delivers hyper-realistic and intelligent digital human interactions. However, these capabilities inherently impose significant computational demands, potentially limiting accessibility for users with resource-constrained devices. To bridge this gap and extend the system’s usability across diverse hardware environments, we have systematically implemented a series of optimization strategies designed to balance computational efficiency with performance fidelity.

(1) *Scalable Cloud-Based Deployment:* To alleviate the computational burden on client devices, we leverage a cloud-based infrastructure to offload heavy processing tasks to high-performance servers. This architecture enables resource-limited devices, such as smartphones or entry-level PCs, to interact with the digital human system by connecting to backend servers for real-time processing. This approach ensures that even devices with modest hardware capabilities can access the system’s full suite of features without compromising responsiveness or accuracy.

(2) *Model Optimization Techniques:* (i) *Quantization:* Neural network weights are compressed through precision reduction (e.g., from 32-bit floating-point to 8-bit integers), significantly lowering memory requirements and accelerating inference speed. This makes the system more computationally viable on edge devices with limited resources. (ii) *Pruning:* Redundant or less significant model parameters are selectively removed via structured pruning, reducing model size and computational complexity while preserving predictive accuracy. (iii) *Knowledge Distillation:* We train smaller but more efficient models to mimic the behavior of larger and more complex models through knowledge distillation. This technique allows the distilled models to retain much of the original model’s performance while drastically reducing computational demands.

(3) *Edge Computing for Low-Latency Processing:* For latency-sensitive applications, we incorporate edge computing strategies, enabling certain tasks, such as ASR, to be processed directly on the user’s device. This reduces round-trip communication with the server, accelerating response time and improving real-time interactivity, even in bandwidth-limited environments.

(4) *Adaptive Rendering for Dynamic Performance Management:* To accommodate devices with varying graphical capabilities, the system dynamically adjusts rendering quality based on available computational resources. Higher-end devices can experience full-fidelity 3D rendering, while lower-end devices receive simplified visual outputs, preserving functional integrity while optimizing performance.

(5) *Efficient Data Transfer Mechanisms:* We implement data compression and batch processing techniques to reduce the volume and frequency of data transmitted between the client and server. This minimizes network load, decreases latency, and conserves bandwidth, making the system more resilient in low-connectivity environments.

These collective strategies ensure that our digital human system remains accessible and practical across a broad spectrum of devices and network conditions. By intelligently distributing computational tasks, streamlining model architecture, and adapting rendering quality, we successfully mitigate the constraints imposed by hardware limitations, democratizing access to state-of-the-art digital human interactions.

## 7. Conclusions

The article presents our award-winning project, a LLM-driven 3D hyper-realistic intelligent digital human system, which demonstrates the potential of AI to revolutionize the production and consumption of digital content. The system utilizes ASR, NLP, and emotional TTS technologies to create a 3D photorealistic digital human interactive environment. The system takes various forms of rich media data as input and produces a digital avatar representation that faithfully captures the essence of the original individual or animal subject. The system is designed with a modular concept and client–server architecture that emphasizes the separation of components for scalable development and efficient progress. The distributed architecture, coupled with advanced deep learning and NLP techniques, empowers the digital workforce to seamlessly adapt to a wide range of application scenarios, including virtual anchors, virtual customer service representatives, and virtual shopping assistants.

## Figures and Tables

**Figure 1 sensors-25-01855-f001:**
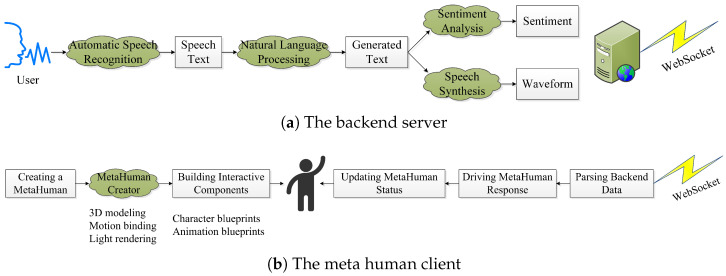
The proposed framework for a large language model-driven 3D hyper-realistic interactive intelligent digital human system.

**Figure 2 sensors-25-01855-f002:**
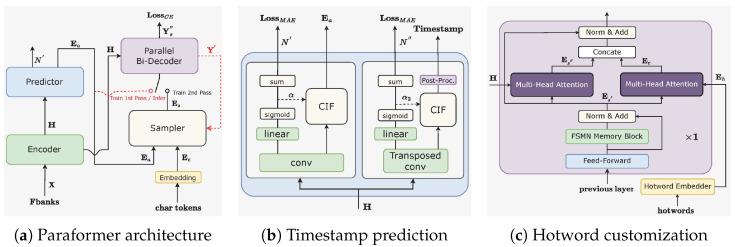
Illustrations of the Paraformer-related architectures.

**Figure 3 sensors-25-01855-f003:**
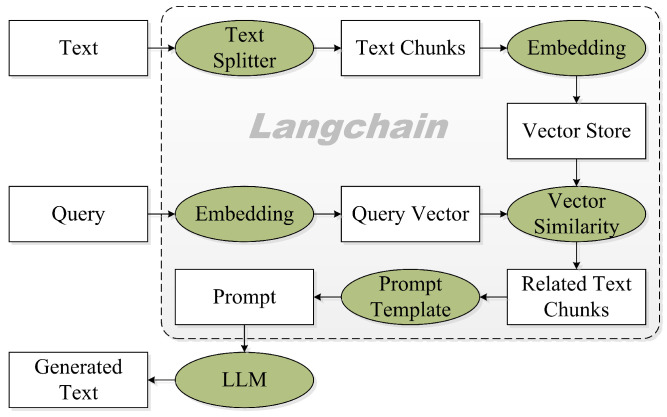
Local knowledge-based LLM (e.g., ChatGLM or ChatGPT) Q/A applications with langchain.

**Figure 4 sensors-25-01855-f004:**
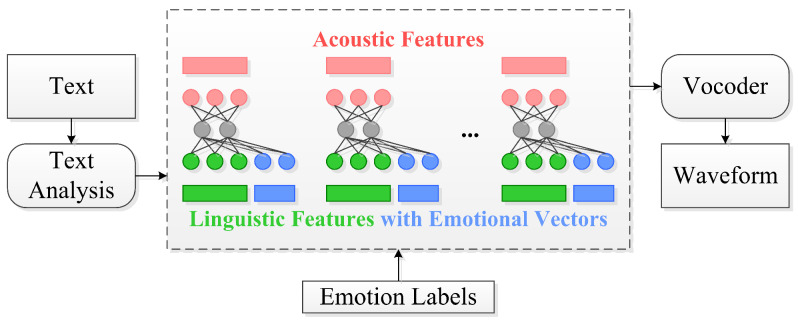
Emotional and expressive TTS.

**Figure 5 sensors-25-01855-f005:**
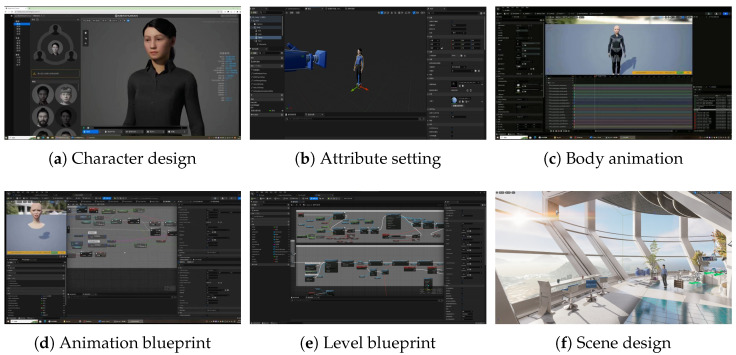
Several screenshots of the design flow for the digital human client.

**Table 1 sensors-25-01855-t001:** Comparison of character error rate (CER) on Aishell-1, Aishell-2, and WenetSpeech benchmark datasets with open-source speech recognition toolkits.

Model	Parameters	Aishell-1	Aishell-2	WenetSpeech
		Test	Test_iOS	Test_Meeting
ESPNET Conformer [9]	46 M	4.60	5.70	15.90
K2 Transducer [10]	80 M	5.05	5.56	14.44
PaddleSpeech DeepSpeech2 [11]	58 M	6.40	–	–
WeNet Conformer-U2++ [13]	47 M	4.40	5.35	17.34
Paraformer	68 M	3.69	4.63	8.32
Paraformer-large	220 M	**1.94**	**2.84**	**7.01**

**Table 2 sensors-25-01855-t002:** Mean win rate of each evaluated foundation language model. For three metrics (accuracy, robustness, and fairness), higher is better; for the remaining three metrics (calibration, bias, and toxicity) lower is better.

Model	Window Size	Accuracy	Calibration	Robustness	Fairness	Bias	Toxicity
Anthropic-LM (52B)	8192	0.780	–	0.818	0.794	0.593	0.649
BLOOM (176B)	2048	0.446	0.348	0.541	0.551	0.546	0.596
Cohere xlarge (52.4B)	2047	0.560	0.543	0.506	0.550	0.598	0.574
J1-Jumbo v1 (178B)	2047	0.517	0.666	0.452	0.488	0.549	0.604
OPT (175B)	2048	0.609	0.338	0.519	0.622	0.580	0.435
T5 (11B)	512	0.131	0.435	0.164	0.150	0.489	0.576
TNLG v2 (530B)	2047	0.787	0.615	0.650	0.752	0.531	0.330
YaLM (100B)	2048	0.075	0.402	0.205	0.167	0.379	0.242
davinci (175B)	2048	0.538	0.575	0.509	0.558	0.445	0.422
text-davinci-002	4000	0.905	0.474	0.916	0.864	0.502	0.409
GLM (130B)	2048	0.512	0.652	0.647	0.513	0.451	0.335

## Data Availability

The data presented in this study are available upon request from the corresponding author. The data are not publicly available due to privacy restrictions.

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
