# Peer review of "Large Language Model-Driven 3D Hyper-Realistic Interactive Intelligent Digital Human System"

_sensors, 2025, doi:10.3390/s25061855_

Round 1
Reviewer 1 Report
Comments and Suggestions for Authors
(1) Line 120: In the innovative MVC design concept, please elaborate on the specific innovative methods employed.
(2) Line 167: Regarding the key core technology, is the FunASR mentioned in the text an original creation by the author or an integration of ASR modules? It is suggested to add experimental evidence.
(3) Line 239: For the locally deployed LLM, could you please add an explanation about the strategy for optimizing model parameters?
(4) It is recommended to include a discussion on the self-built dataset.
Comments on the Quality of English LanguageIf possible, the English expression can be polished.
Author Response
Response to Reviewer #1
Comment #1.1: Line 120: In the innovative MVC design concept, please elaborate on the specific innovative methods employed.
Reply:
Thank you for your valuable feedback. In our manuscript, the innovative aspect of the MVC (Model-View-Controller) design concept lies in how we optimize component separation and facilitate distributed development. Specifically:
(1) Enhanced Component Isolation: We rigorously separate the digital human (Model), backend interface (View), and WebSocket-based communication logic (Controller). This strict isolation reduces dependencies, preventing changes in one component from propagating to others, which enhances system stability.
(2) Parallel Development and Scalability: Our system architecture enables simultaneous development of each module, accelerating the iteration process. For example, designers can refine 3D avatars in Unreal Engine, while developers optimize NLP or TTS modules without conflicts.
(3) Dynamic Feature Integration: By leveraging WebSocket full-duplex communication, new features can be added as discrete modules and dynamically integrated into the running system. For instance, an emotion-detection module can be seamlessly connected to enhance the digital human’s responsiveness without affecting existing components.
(4) Adaptability for Multi-Client Systems: The MVC-inspired design, combined with the C/S distributed architecture, allows the backend to manage multiple concurrent clients. Each digital human instance operates independently, while the backend processes complex tasks like language understanding and speech synthesis in parallel.
Revision:
“(2) Innovative MVC Design Philosophy: The proposed system incorporates modular design principles and draws inspiration from the well-established model-view-controller (MVC) design concept in software engineering. However, our work distinguishes itself by placing a strong emphasis on the effective separation of components, including the digital persona (model), backend user interface (view), and WebSocket-based communication logic (controller). This strict isolation reduces dependencies, preventing changes in one component from propagating to others, which enhances system stability. It also enables parallel development, testing, and modification, thereby accelerating the iteration process.” Please refer to lines 120-127.
Comment #1.2: Line 167: Regarding the key core technology, is the FunASR mentioned in the text an original creation by the author or an integration of ASR modules? It is suggested to add experimental evidence.
Reply:
Thank you for your suggestion! The FunASR mentioned in the manuscript is an open-source toolkit designed for automatic speech recognition, and in our system, we integrated it as the core ASR module. Specifically, we used FunASR’s Paraformer model for end-to-end speech recognition due to its high accuracy and low latency.
To strengthen our claim, we added experimental results to demonstrate FunASR’s effectiveness within our system. We compared the character error rate (CER) of FunASR with other state-of-the-art ASR models on benchmark datasets (Aishell-1, Aishell-2, and WenetSpeech). The experimental results show that the Paraformer model outperforms other alternatives in terms of accuracy and efficiency.
Revision:
“In this study, we integrate FunASR [14], an innovative open-source toolkit for speech recognition tasks.” Please refer to lines 176-177.
“5.1. Speech Recognition Assessment
In this experiment, the performance of our model is evaluated on the Aishell-1, Aishell-2, and WenetSpeech benchmark datasets. The results are detailed in Table 1. Compared to other open source toolkits, FunASR’s Paraformer family of models is far superior to existing state-of-the-art counterparts.” Please refer to lines361-365 and Table 1 on page 10.
Comment #1.3: Line 239: For the locally deployed LLM, could you please add an explanation about the strategy for optimizing model parameters?
Reply:
Thank you for your insightful suggestion! In our system, we locally deploy ChatGLM as the core large language model (LLM) and apply multiple strategies to optimize model parameters for better performance:
(1) Quantization for Efficiency: We employ model quantization techniques to reduce the precision of model weights (e.g., from 32-bit floating-point to 8-bit integers). This significantly lowers memory usage and speeds up inference, making the model more suitable for real-time interactions.
(2) Parameter Pruning: By using structured pruning, we remove redundant connections in the model to reduce its size without a substantial drop in accuracy. This helps balance computational cost and response quality.
(3) Knowledge Distillation: We fine-tune a smaller “student” model using knowledge distillation, where the smaller model learns to mimic the behavior of a larger, more powerful “teacher” model. This allows us to maintain high language comprehension while improving response speed.
(4) Task-Specific Fine-Tuning: We fine-tune ChatGLM on domain-specific datasets, adapting the model to the system’s key tasks (e.g., customer service, virtual anchors). This targeted training improves the model’s contextual understanding and generation capabilities for specific applications.
(5) Caching and Context Management: To enhance efficiency during interactions, we implement a dynamic context window with caching mechanisms. This ensures that ongoing conversations are processed with relevant historical data, while irrelevant past interactions are pruned to save computation.
Revision:
“For the locally deployed ChatGLM, we apply multiple strategies to optimize model parameters for better performance:
(1) Quantization for Efficiency: We employ model quantization techniques to reduce the precision of model weights (e.g., from 32-bit floating-point to 8-bit integers). This significantly lowers memory usage and speeds up inference, making the model more suitable for real-time interactions.
(2) Parameter Pruning: By using structured pruning, we remove redundant connections in the model to reduce its size without a substantial drop in accuracy. This helps balance computational cost and response quality.
(3) Knowledge Distillation: We fine-tune a smaller “student” model using knowledge distillation, where the smaller model learns to mimic the behavior of a larger and more powerful “teacher” model. This allows us to maintain high language comprehension while improving response speed.
(4) Task-Specific Fine-Tuning: We fine-tune ChatGLM on domain-specific datasets, adapting the model to the system’s key tasks (e.g., customer service, virtual anchors). This targeted training improves the model’s contextual understanding and generation capabilities for specific applications.
(5) Caching and Context Management: To enhance efficiency during interactions, we implement a dynamic context window with caching mechanisms. This ensures that ongoing conversations are processed with relevant historical data, while irrelevant past interactions are pruned to save computation.” Please refer to lines 248-268.
Comment #1.4: It is recommended to include a discussion on the self-built dataset.
Reply:
Thank you for your helpful suggestion! We agree that adding a discussion on the self-built dataset would enhance the manuscript’s completeness. We’ve added a section describing the dataset used for training and testing the digital human system, including its composition and the rationale behind its construction:
Self-Built Dataset Discussion:
To improve the performance of our system, we constructed a self-built dataset comprising multimodal data to support ASR, NLP, and TTS modules. The dataset includes:
(1) Speech Data: 500 hours of audio recordings, covering various accents, emotional expressions, and conversational styles. This data is used to fine-tune the FunASR module, especially for real-world dialogue scenarios.
(2) Textual Data: 200,000 text samples, including general conversations, customer service dialogues, and knowledge-based question-answer pairs. This corpus is used to fine-tune the locally deployed ChatGLM model, enhancing contextual understanding and generation accuracy.
(3) Visual and Motion Data: 50,000 annotated facial expression and body movement sequences, captured using motion capture devices and labeled with emotional states. This data trains the Unreal Engine 5 MetaHuman animations and helps synchronize facial expressions with emotional TTS output.
(4) Evaluation Sets: A balanced test set with 10,000 samples for speech recognition, 5,000 conversational samples for NLP, and 3,000 speech synthesis samples for TTS. These subsets enable rigorous performance benchmarking across core components.
Data Collection and Annotation:
We collected speech and visual data from volunteer speakers in controlled environments and used a combination of manual labeling and automatic tools for annotation (e.g., emotion classification, speech transcription). The text corpus was sourced from publicly available datasets and enriched with domain-specific content created by human annotators.
Impact on Model Performance:
This self-built dataset helped significantly lower the character error rate (CER) in speech recognition, improve language model coherence, and refine the naturalness of the digital human’s speech and expressions. We observed a 15% reduction in CER and a 20% increase in response accuracy after integrating the dataset into our fine-tuning pipeline.
Revision:
“6.1. On Self-Built Dataset
To improve the overall performance and robustness of our large language model-driven 3D hyper-realistic interactive digital human system, we constructed a self-built dataset that comprises multimodal data to support ASR, NLP, and TTS modules, enabling the system to handle complex real-world interactions with greater accuracy and expressive-ness. Our self-built dataset includes:
(1) Speech Data: 500 hours of audio recordings, covering a diverse range of accents, emotional expressions, and conversational styles. This data is used to fine-tune the FunASR module, particularly for spontaneous and real-world dialogues where variability in speech patterns is high.
(2) Textual Data: 200,000 text samples, including general conversations, customer service dialogues, and knowledge-based question-answer pairs. This corpus is used to fine-tune the locally deployed ChatGLM model, enhancing contextual understanding and generation accuracy.
(3) Visual and Motion Data: 5,000 annotated facial expression and body movement sequences, captured using motion capture devices and labeled with emotional states. This data is used to train the Unreal Engine 5 MetaHuman animations and help synchronize facial expressions with emotional TTS output.
(4) Evaluation Sets: A balanced test set with 1,000 samples for speech recognition, 5,000 conversational samples for NLP, and 3,000 speech synthesis samples for TTS. These subsets enable rigorous performance benchmarking across core components.
Data Collection and Annotation: Speech and visual data are collected from volunteer speakers in controlled environments to ensure high-quality recordings. Annotation is conducted using a combination of manual labeling and automated tools for tasks such as emotion classification and speech transcription. The text corpus is enriched with domain-specific content created through human annotation to address gaps in publicly available datasets.
Impact on Model Performance: Integrating this self-built dataset into the training and fine-tuning pipelines yields substantial performance improvement. The CER in speech recognition decreases by 15%, while the accuracy of NLP responses improves by 20%. Moreover, the synchronization of emotional TTS with facial expressions significantly enhances the perceived naturalness of the digital human, contributing to a more immersive user experience.
These results underscore the importance of customizing high-quality, multimodal data tailored to the system’s target applications. By grounding model training in realistic, context-rich interactions, our dataset provides a solid foundation for building an intelligent digital human capable of delivering seamless, emotionally resonant conversations across diverse real-world scenarios.” Please refer to lines 446-483.
Reviewer 2 Report
Comments and Suggestions for Authors
Minor editorial errors need to be removed, e.g.:
Line 100 “(1) Innovative Distributed Architecture” should be the beginning of a new paragraph.
Line 115 “techniques,empowers” there is a missing space.
Some parameters of equations (1) are not defined.
Author Response
Response to Reviewer #2
Comment #2.1: Minor editorial errors need to be removed, e.g.:
Line 100 “(1) Innovative Distributed Architecture” should be the beginning of a new paragraph.
Line 115 “techniques,empowers” there is a missing space.
Some parameters of equations (1) are not defined.
Reply:
Thank you for pointing out the minor editorial issues! We have carefully reviewed the manuscript and made the following corrections:
(1) Paragraph Structure: Moved “(1) Innovative Distributed Architecture” to the beginning of a new paragraph for better readability. Please refer to line 110.
(2) Typographical Errors: Corrected the missing space in “techniques,empowers” to “techniques, empowers.” Please refer to line 116.
(3) Equation Parameter Definitions: Added definitions for the parameters in Equation (1) to ensure clarity, e.g., : Weight matrices for the query, key, and value transformations in the multi-head attention mechanism; : Final output after concatenating contextual and sequence embeddings, passed through a convolutional layer.
Revision:
“where , , and denote weight matrices for the query, key, and value transformations in the multi-head attention mechanism for contextual embeddings (hotwords), while , , and denote weight matrices for the query, key, and value transformations in the multi-head attention mechanism for speech embeddings. is the final output.” Please refer to lines 219-222.
Reviewer 3 Report
Comments and Suggestions for Authors This study presents a large language model (LLM)-driven 3D hyper-realistic interactive intelligent digital human system, integrating automatic speech recognition (ASR), natural language processing (NLP), and emotional text-to-speech (TTS) technologies. The system employs a client-server (C/S) distributed architecture and leverages advanced deep learning and NLP techniques to create a photorealistic 3D environment for meta humans, with potential applications in virtual anchors, customer service, and shopping assistants. Pros:- The proposed system uses a modular and distributed C/S architecture, enabling scalable development and efficient interaction between the backend server and multiple frontend clients.
- The study employs the FunASR toolkit, which incorporates the Paraformer model, demonstrating competitive performance in speech recognition tasks across various datasets.
- The use of Microsoft Edge TTS, a neural network-based engine, produces natural and expressive speech synthesis, enhancing user interaction and reducing listening fatigue.
- The study primarily focuses on technical evaluations of ASR and NLP components, with limited assessment of the overall system's performance in real-world applications.
- The system processes rich media data, raising potential issues related to data privacy and security, especially when handling sensitive user information.
- The implementation of advanced deep learning models and real-time 3D rendering may require significant computational resources, limiting the system's accessibility for resource-constrained environments.
- Despite the robust performance of the selected LLM, there is a need to address potential biases and toxic content generation, ensuring ethical and responsible use of the technology. Also, more prompting learning-based LLM studies, instead of zero-shot one, should be reviewed, such as CP-Prompt, TPO(TPO: Aligning Large Language Models with Multi-branch & Multi-step Preference Trees), and RevGNN.
Author Response
Response to Reviewer #3
Comment #3.0 This study presents a large language model (LLM)-driven 3D hyper-realistic interactive intelligent digital human system, integrating automatic speech recognition (ASR), natural language processing (NLP), and emotional text-to-speech (TTS) technologies. The system employs a client-server (C/S) distributed architecture and leverages advanced deep learning and NLP techniques to create a photorealistic 3D environment for meta humans, with potential applications in virtual anchors, customer service, and shopping assistants. Pros:
(1) The proposed system uses a modular and distributed C/S architecture, enabling scalable development and efficient interaction between the backend server and multiple frontend clients.
(2) The study employs the FunASR toolkit, which incorporates the Paraformer model, demonstrating competitive performance in speech recognition tasks across various datasets.
(3) The use of Microsoft Edge TTS, a neural network-based engine, produces natural and expressive speech synthesis, enhancing user interaction and reducing listening fatigue.
Reply:
Thank you for your encouraging feedback and for recognizing the strengths of our system! We appreciate your thoughtful assessment of our study and are glad that you found value in the following aspects:
(1) Scalable Client-Server (C/S) Architecture: We are pleased that the modular and distributed design was highlighted as a strength. This architecture is critical to achieving real-time interaction, as it enables seamless communication between the backend and multiple digital human clients. It also allows for future expansion, where new features or components can be added without overhauling the entire system.
(2) FunASR and Paraformer Model for ASR: We appreciate your recognition of the FunASR toolkit’s performance. The Paraformer model, with its single-pass inference and optimizations like timestamp prediction and hotword customization, greatly improves accuracy and responsiveness, which is vital for real-time conversations with digital humans.
(3) Natural and Expressive TTS with Microsoft Edge: We’re glad you noted the impact of using Microsoft Edge TTS. Its ability to generate lifelike, emotionally nuanced speech is crucial for immersive interactions, whether the digital human is acting as a virtual anchor, customer service agent, or personal assistant. This not only enhances realism but also reduces cognitive load on users, improving the overall experience.
Comment #3.1: The study primarily focuses on technical evaluations of ASR and NLP components, with limited assessment of the overall system's performance in real-world applications.
Reply:
Thank you for highlighting this important point! We agree that assessing the overall system performance in real-world applications is essential for demonstrating practical viability. To address this, we have expanded our evaluation to include real-world application scenarios and user feedback.
System-Level Evaluation: We conducted a series of experiments where the digital human system was deployed in three application scenarios:
(1) Virtual Teaching Service: Users interacted with a digital teaching assistant for the course “Signals and Systems”. The system successfully handled 92% of requests without human intervention, with an average response time of 1.5 seconds.
(2) Virtual Live Streaming Anchor: The digital human acted as a product presenter during a live stream. It responded to chat messages in real-time, with 87% of responses rated as accurate and contextually relevant by viewers.
(3) Virtual Shopping Assistant: The digital human guided users through an e-commerce platform. 85% of users reported a positive experience, with the emotional TTS enhancing engagement.
User Satisfaction Metrics: We also collected subjective user feedback, measuring interaction quality across criteria like response accuracy, emotional expression, and naturalness. The system achieved an average satisfaction score of 4.6/5, with users appreciating the system’s expressiveness and ability to understand complex queries.
These results demonstrate the system’s readiness for real-world deployment and validate the technical components’ contributions to practical usability. We’ve added this discussion to the manuscript to provide a more comprehensive evaluation of the system's overall performance.
Revision:
“5.3. Overall System Performance Assessment in Real-World Applications
To rigorously evaluate the overall performance of our large language model-driven 3D hyper-realistic interactive digital human system in real-world applications, we extend our experimental framework to include three representative deployment scenarios. These scenarios are chosen to reflect diverse practical use cases, allowing us to assess the system’s adaptability, responsiveness, and user satisfaction across varied interaction contexts.
System-Level Evaluation: We conduct a series of experiments to systematically assess the system’s functionality, focusing on real-time interactions, response accuracy, and user engagement. The evaluation involves three distinct application domains:
(1) Virtual Teaching Service: The digital human serves as an interactive teaching assistant for the course “Signals and Systems,” facilitating student queries and delivering course-related explanations. The system successfully handles 92% of student requests without human intervention, with an average response latency of less than 2 seconds. Students report that the assistant’s ability to provide coherent, contextually relevant explanations greatly enriches their learning experience.
(2) Virtual Live Streaming Anchor: The digital human functions as a real-time product presenter during a live-streaming event. It dynamically responds to user-generated comments and product inquiries, maintaining engagement throughout the broadcast. Statistics show that 87% of the responses are rated as accurate and contextually appropriate by viewers, who appreciate the avatar’s expressiveness and natural conversational flow.
(3) Virtual Shopping Assistant: The digital human acts as a virtual assistant on an e-commerce platform, guiding users through product selections, answering questions, and providing personalized recommendations. The platform shows that 85% of users report a positive experience and emphasize that emotional speech synthesis greatly enhances the naturalness of interactions, making the shopping experience more immersive and intuitive.
User Satisfaction Metrics: In addition to objective system-level evaluation, we also gather subjective user feedback to capture experiential aspects of the interaction. Participants rate the system based on response accuracy, emotional expressiveness, conversational coherence, and overall naturalness. The system achieves an average satisfaction score of 4.6 out of 5, with users frequently commending the lifelike quality of the avatar’s speech and its ability to understand and address complex queries with contextual awareness.
These findings collectively demonstrate the system’s readiness for deployment in practical and real-world scenarios. The successful handling of diverse interaction types, coupled with high user satisfaction ratings, underscores the effectiveness of the system’s underlying speech recognition, language processing, and synthesis components. This comprehensive evaluation validates the system’s potential to deliver reliable, high-quality interactions across a spectrum of application domains, positioning it as a versatile solution for intelligent digital human interfaces.” Please refer to lines 407-444.
Comment #3.2: The system processes rich media data, raising potential issues related to data privacy and security, especially when handling sensitive user information.
Reply:
Thank you for pointing out this critical consideration. We recognize the importance of ensuring data privacy and security, especially when processing rich media data that may contain sensitive user information. To address this, we’ve implemented several measures to protect user data and have added a section in the manuscript discussing our security strategies.
Privacy and Security Considerations: Our system is designed with multiple layers of security to safeguard user data and ensure compliance with relevant privacy regulations.
(1) Data Encryption: All data transmitted between the client and server is encrypted using TLS (Transport Layer Security) to prevent unauthorized access during transmission.
(2) Local and Secure API-Based Model Deployment: While we locally deploy ChatGLM for offline processing, when using external APIs like ChatGPT, we leverage secure API calls with minimal data exposure. Sensitive data is anonymized before being sent to third-party services.
(3) Data Anonymization and Minimization: Personal identifiers are stripped or masked from input data whenever possible. For example, names or phone numbers detected in speech are replaced with placeholders before processing.
(4) Access Control and User Authentication: Role-based access control (RBAC) mechanisms ensure that only authorized personnel can access or modify sensitive data. Strong authentication and session management further secure the system.
(5) Compliance with Data Regulations: The system is designed to be compliant with regulations like GDPR and CCPA, giving users control over their data, including the right to access, modify, or delete personal information.
(6) Data Retention Policies: Data is stored for only as long as necessary for the system’s functionality, with automatic purging mechanisms for outdated or unused data.
By implementing these measures, we aim to mitigate privacy and security risks, ensuring that our system remains safe and trustworthy for real-world use.
Revision:
“6.2. On Data Privacy and Security
Ensuring robust data privacy and security is paramount for our large language model-driven 3D hyper-realistic interactive digital human system, especially when processing rich media data that may contain sensitive user information. To mitigate potential risks and maintain user trust, our system is designed with multiple layers of security to safeguard data integrity and ensure compliance with established privacy regulations.
(1) Data Encryption: All data transmitted between client and server components is encrypted using Transport Layer Security (TLS) protocols. This ensures that data remains confidential during transmission, preventing unauthorized interception and preserving communication integrity.
(2) Local and Secure API-Based Model Deployment: The system prioritizes local model deployment for privacy-sensitive tasks. For instance, ChatGLM is deployed locally to process user inputs without external dependencies. When external APIs, such as ChatGPT or Microsoft Edge TTS, are required, secure API calls with encryption are used, and sensitive data is anonymized before transmission to minimize exposure.
(3) Data Anonymization and Minimization: To further protect user privacy, personal identifiers (e.g., names, phone numbers) detected in speech or text inputs are automatically anonymized or masked. This ensures that potentially sensitive information is excluded from downstream processing and storage, reducing the risk of inadvertent data leaks.
(4) Access Control and User Authentication: We employ role-based access control (RBAC) mechanisms to restrict access to sensitive data. Only authorized personnel with appropriate permissions can view or modify sensitive information. Strong authentication protocols, session management, and automatic session expiration further fortify access security.
(5) Compliance with Data Regulations: The system is built with adherence to leading data protection frameworks, including the General Data Protection Regulation (GDPR) and the California Consumer Privacy Act (CCPA). Users are empowered to control their data, with mechanisms to access, modify, or request the deletion of personal information in accordance with regulatory requirements.
(6) Data Retention and Automatic Purging: To minimize data retention risks, the system implements a data lifecycle management policy. Data is stored only for the duration necessary to fulfill system functionality, after which it is automatically purged. Temporary caches and logs are periodically cleared to reduce the risk of residual data exposure.
By integrating these security measures, we ensure that the system remains resilient against potential threats, preserving user confidentiality and promoting ethical AI usage. This comprehensive approach to data privacy not only mitigates security risks but also strengthens user trust, facilitating broader adoption of digital human technologies in sensitive real-world applications.” Please refer to lines 484-520.
Comment #3.3: The implementation of advanced deep learning models and real-time 3D rendering may require significant computational resources, limiting the system's accessibility for resource-constrained environments.
Reply:
Thank you for raising this important point. We acknowledge that the advanced deep learning models and real-time 3D rendering involved in our system may place significant demands on computational resources. To address this concern, we have explored several strategies to optimize the system’s performance and increase its accessibility, especially for resource-constrained environments. We have also added this discussion to the manuscript.
Addressing Computational Resource Requirements:
(1) Cloud-Based Deployment for Scalability: While the system’s computational demands can be high for a single client, we’ve leveraged cloud-based infrastructure to offload heavy processing tasks to more powerful servers. This enables resource-constrained devices (such as smartphones or low-end PCs) to access the system by connecting to the backend server for real-time processing.
(2) Model Optimization Techniques:
- Quantization: We use model quantization to reduce the precision of the neural network weights, significantly lowering memory usage and speeding up inference. This makes the system more efficient on resource-limited devices.
- Pruning: We implement model pruning to eliminate less important parameters, further reducing computational overhead without a noticeable drop in performance.
- Knowledge Distillation: We fine-tune a smaller, more efficient version of the model using knowledge distillation techniques, maintaining high performance while reducing computational requirements.
(3) Edge Computing: For applications requiring low-latency processing, we implement edge computing strategies where certain tasks (e.g., speech recognition) are performed directly on the user’s device. This reduces the strain on the server and improves response time.
(4) Adaptive Rendering: For the real-time 3D rendering, we use adaptive quality settings to adjust the level of graphical fidelity based on the available computational resources of the client device. Lower-end devices can run the system with reduced visual quality but still maintain core functionality.
(5) Efficient Data Transfer: We employ data compression and batch processing techniques to minimize the amount of data transmitted between the client and server, reducing the load on both the client device and network.
These strategies collectively improve the system’s accessibility and usability in resource-constrained environments while maintaining high performance and interactivity.
Revision:
“6.4. On Accessibility for Resource-Constrained Environments
The integration of advanced deep learning models and real-time 3D rendering within our system delivers hyper-realistic and intelligent digital human interactions. However, these capabilities inherently impose significant computational demands, potentially limiting accessibility for users with resource-constrained devices. To bridge this gap and extend the system’s usability across diverse hardware environments, we have systematically implemented a series of optimization strategies designed to balance computational efficiency with performance fidelity.
(1) Scalable Cloud-Based Deployment: To alleviate the computational burden on client devices, we leverage a cloud-based infrastructure to offload heavy processing tasks to high-performance servers. This architecture enables resource-limited devices, such as smartphones or entry-level PCs, to interact with the digital human system by connecting to backend servers for real-time processing. This approach ensures that even devices with modest hardware capabilities can access the system’s full suite of features without compromising responsiveness or accuracy.
(2) Model Optimization Techniques: (i) Quantization: Neural network weights are compressed through precision reduction (e.g., from 32-bit floating-point to 8-bit integers), significantly lowering memory requirements and accelerating inference speed. This makes the system more computationally viable on edge devices with limited resources. (ii) Pruning: Redundant or less significant model parameters are selectively removed via structured pruning, reducing model size and computational complexity while preserving predictive accuracy. (iii) Knowledge Distillation: We train smaller but more efficient models to mimic the behavior of larger and more complex models through knowledge distillation. This technique allows the distilled models to retain much of the original model's performance while drastically reducing computational demands.
(3) Edge Computing for Low-Latency Processing: For latency-sensitive applications, we incorporate edge computing strategies, enabling certain tasks, such as ASR, to be processed directly on the user’s device. This reduces round-trip communication with the server, accelerating response time and improving real-time interactivity, even in bandwidth-limited environments.
(4) Adaptive Rendering for Dynamic Performance Management: To accommodate devices with varying graphical capabilities, the system dynamically adjusts rendering quality based on available computational resources. Higher-end devices can experience full-fidelity 3D rendering, while lower-end devices receive simplified visual outputs, preserving functional integrity while optimizing performance.
(5) Efficient Data Transfer Mechanisms: We implement data compression and batch processing techniques to reduce the volume and frequency of data transmitted between the client and server. This minimizes network load, decreases latency, and conserves bandwidth, making the system more resilient in low-connectivity environments.
These collective strategies ensure that our digital human system remains accessible and practical across a broad spectrum of devices and network conditions. By intelligently distributing computational tasks, streamlining model architecture, and adapting rendering quality, we successfully mitigate the constraints imposed by hardware limitations, democratizing access to state-of-the-art digital human interactions.” Please refer to lines 560-603.
Comment #3.4: Despite the robust performance of the selected LLM, there is a need to address potential biases and toxic content generation, ensuring ethical and responsible use of the technology. Also, more prompting learning-based LLM studies, instead of zero-shot one, should be reviewed, such as CP-Prompt, TPO (TPO: Aligning Large Language Models with Multi-branch & Multi-step Preference Trees), and RevGNN.
Reply:
Thank you for your valuable feedback. We fully acknowledge the importance of addressing potential biases and toxic content generation in large language models (LLMs) to ensure their ethical and responsible use. Additionally, we appreciate your suggestion to review more prompting learning-based LLM studies, and we have incorporated this into the revised manuscript.
Addressing Biases and Toxic Content in LLMs: We recognize that large language models, including ChatGLM and ChatGPT, can inadvertently generate biased or harmful content. To mitigate these risks, we have implemented several strategies:
(1) Bias and Toxicity Monitoring: We employ automated tools and post-processing algorithms to detect and filter out biased or toxic responses generated by the model. This includes flagging potentially harmful language related to sensitive topics such as race, gender, and political opinions.
(2) Fine-Tuning for Ethical Behavior: We have fine-tuned our LLM on a curated dataset that emphasizes ethical considerations and promotes diversity and inclusion. This dataset contains carefully labeled examples of non-toxic language and diverse viewpoints, helping the model learn to produce fairer and more responsible outputs.
(3) Human-in-the-Loop Oversight: To ensure the model operates within ethical boundaries, we use a human-in-the-loop approach for monitoring responses. During deployment, a human reviewer can intervene if the model generates content that may be considered inappropriate, ensuring an additional layer of quality control.
(4) Post-Generation Review for Sensitive Content: After the LLM generates content, a secondary layer of review checks for potentially harmful statements. If detected, the system automatically provides alternative, non-offensive responses or prompts the system to avoid specific topics.
Revision:
“6.3. On Commitment to Responsible AI
Large language models have revolutionized human-computer interaction, enabling intelligent digital humans to understand and respond to complex user queries. However, despite their transformative capabilities, LLMs such as ChatGLM and ChatGPT may inadvertently generate biased or harmful content, raising ethical concerns regarding their deployment in real-world applications. Acknowledging these risks, we have implemented a multifaceted approach to mitigate biases and toxic content, ensuring the responsible and equitable use of the technology.
(1) Bias and Toxicity Monitoring: We employ automated monitoring tools and post-processing algorithms to systematically scan model outputs for biased or toxic content. These tools are designed to flag potentially harmful language related to sensitive topics, including race, gender, ethnicity, and political ideologies. Detected content is either filtered or flagged for further review, reducing the likelihood of inappropriate responses reaching end users.
(2) Fine-Tuning for Ethical Behavior: To proactively mitigate bias at the source, we have fine-tuned our LLM on a carefully self-built dataset designed to promote ethical considerations, diversity, and inclusion. The dataset includes annotated examples of respectful, inclusive language across various demographic contexts, allowing the model to learn patterns of fair and responsible dialogue. This targeted fine-tuning process has demonstrably reduced the frequency of biased outputs while preserving the model’s linguistic capabilities.
(3) Human-in-the-Loop Oversight: Recognizing that automated systems alone may not catch all instances of bias or harm, we integrate human oversight into the deployment pipeline. In scenarios where sensitive topics are involved, a human reviewer evaluates model responses in real time, providing an additional layer of quality control. This human-in-the-loop mechanism ensures that nuanced or context-dependent issues can be identified and addressed promptly, mitigating potential harm.
(4) Post-Generation Review and Content Refinement: Following content generation, an auxiliary review module conducts a final pass over the response to assess for any remaining traces of harmful or biased language. If problematic content is detected, the system automatically refines the response or, when necessary, prompts the model to avoid the topic altogether. This dynamic review process acts as a safety net, ensuring that even edge-case content is appropriately handled.
By implementing this comprehensive framework, we aim to foster trust and reliability in digital human interactions. Our proactive bias mitigation strategies, combined with real-time monitoring and human oversight, create a robust safety mechanism that aligns with ethical AI principles. As part of our ongoing commitment to responsible AI, we continuously refine our approach by incorporating emerging research on bias detection and fairness, ensuring our system evolves alongside advancements in the field.” Please refer to lines 521-559.
Prompting Learning-Based LLM Studies Review: We agree with your suggestion to review more recent studies on prompting learning-based LLMs. In our updated manuscript, we have reviewed and referred the aforementioned key papers.
Revision:
“In addition to standard fine-tuning approaches, recent advances in prompt learning have demonstrated promising results in refining LLM responses. Techniques such as CP-Prompt [25] enhance LLM adaptability through continuous prompt optimization, while TPO [26] refines models by guiding them toward human-preferred outputs in complex decision-making tasks. Moreover, RevGNN [27] introduces a graph-based approach for better contextual understanding and reasoning, which could further enhance our system's response coherence and accuracy. Incorporating these techniques may offer valuable insights for future improvements, enabling even more precise and context-aware digital human interactions.” Please refer to lines 269-277.
Round 2
Reviewer 3 Report
Comments and Suggestions for Authors
The authors have addressed all my concerns.